# LEARNING CONTROL POLICIES FOR REGION STABILIZATION IN STOCHASTIC SYSTEMS

## ABSTRACT

We consider the problem of learning control policies in stochastic systems which guarantee that the system stabilizes within some specified stabilization region with probability 1. Our approach is based on the novel notion of stabilizing ranking supermartingales (sRSMs) that we introduce in this work. Our sRSMs overcome the limitation of methods proposed in previous works whose applicability is restricted to systems in which the stabilizing region cannot be left once entered under any control policy. We present a learning procedure that learns a control policy together with an sRSM that formally certifies probability-1 stability, both learned as neural networks. Our experimental evaluation shows that our learning procedure can successfully learn provably stabilizing policies in practice.

## 1 INTRODUCTION

Machine learning methods present a promising approach to solving non-linear control problems. However, the key challenge for their deployment in real-world scenarios is that they do not consider hard safety constraints. For instance, the main objective of reinforcement learning (RL) is to maximize expected reward (Sutton & Barto, 2018), but doing this provides no guarantees of the system's safety. This is particularly concerning for safety-critical applications such as autonomous driving or healthcare, in which unsafe behavior of the system might have fatal consequences. Thus, a fundamental challenge for deploying learning-based methods in safety-critical applications such as robotics problems is *formally certifying* safety of learned control policies (Amodei et al., 2016; García & Fernández, 2015).

Stability is a fundamental safety constraint in control theory, which requires the system to converge to and eventually stay within some specified stabilizing region with probability 1, a.k.a. almost-sure (a.s.) asymptotic stability (Khalil, 2002; Kushner, 1965). Most existing research on learning policies for a control system with formal guarantees on stability considers *deterministic* systems and employs Lyapunov functions (Khalil, 2002) for certifying the system's stability. In particular, a Lyapunov function is learned jointly with the control policy (Berkenkamp et al., 2017; Richards et al., 2018; Chang et al., 2019; Abate et al., 2021a). Informally, a Lyapunov function is a function that maps system states to nonnegative real numbers whose value decreases after every one-step evolution of the system until the stabilizing region is reached. Recent work Lechner et al. (2022) has extended the notion of Lyapunov functions to *stochastic* systems and proposed *ranking supermartingales (RSMs)* for certifying a.s. asymptotic stability in stochastic systems. RSMs generalize Lyapunov functions to supermartingale processes in probability theory (Williams, 1991) and decrease in value in *expectation* upon every one-step evolution of the system.

While these works present significant advances in learning control policies with formal stability guarantees, they are either only applicable to deterministic systems or assume that the stabilizing set is *closed under system dynamics*, i.e., the agent cannot leave it once entered. In particular, the work of Lechner et al. (2022) reduces stability in stochastic systems to an *a.s. reachability* condition by assuming that the agent cannot leave the stabilization set. However, this assumption may not hold in real-world settings because the agent may be able to leave the stabilizing set with some positive probability due to the existence of stochastic disturbances. We illustrate this on an example in Figure 1.

**Contributions** In this work, we introduce *stabilizing ranking supermartingales (sRSMs)* and prove that they certify a.s. asymptotic stability even when the stabilizing set is not assumed to be closed

under system dynamics. The key novelty of our sRSMs compared to RSMs is that they also impose an expected decrease condition within a part of the stabilizing region. The additional condition ensures that, once entered, the agent leaves the stabilizing region with a probability at most $p < 1$. Thus, the probability of the agent entering and leaving the stabilizing region $N$ times is at most $p^N$, which by letting $N \to \infty$ implies that the agent eventually stabilizes within the region with probability 1. The key conceptual novelty is that we combine the convergence results of RSMs Lechner et al. (2022) with a *concentration bound* on the supremum value of a supermartingale process. This combined reasoning allows us to formally guarantee a.s. asymptotic stability even for systems in which the stabilizing region is not closed under system dynamics.

We also present a method for learning a control policy jointly with an sRSM that certifies a.s. asymptotic stability. The method parametrizes both the policy and the sRSM as neural networks and draws insight from established procedures for learning neural network Lyapunov functions Chang et al. (2019) and RSMs Lechner et al. (2022). It loops between a learner module that jointly trains a policy and an sRSM candidate and a verifier module that certifies a.s. asymptotic stability of the learned sRSM candidate by formally checking whether all sRSM conditions are satisfied. If the sRSM candidate violates some sRSM conditions, the verifier module produces counterexamples that are added to the learner module's training set to guide the learner in the next loop iteration.

We experimentally evaluate our learning procedure on 2 stochastic RL tasks in which the stabilizing region is not closed under system dynamics and show that our learning procedure successfully learns control policies with a.s. asymptotic stability guarantees for both tasks.

## 2 RELATED WORK

**Stability for deterministic systems** Most early works on control with stability constraints rely either on hand-designed certificates or their computation via sum-of-squares (SOS) programming (Henrion & Garulli, 2005; Parrilo, 2000). Automation via SOS programming is restricted to problems with polynomial dynamics and does not scale well with dimension. Learning-based methods present a promising approach to overcome these limitations (Richards et al., 2018; Jin et al., 2020; Chang & Gao, 2021). In particular, the methods of (Chang et al., 2019; Abate et al., 2021a) also learn a control policy and a Lyapunov function as neural networks by using a learner-verifier framework that our method builds on and extends to stochastic systems.

**Stability for stochastic systems** While the theory behind stochastic system stability is well studied (Kushner, 1965; 2014), there are only a few works that consider control with formal stability guarantees. The methods of (Crespo & Sun, 2003; Vaidya, 2015) are numerical and certify weaker notions of stability. Recently, (Lechner et al., 2022; Žikelić et al., 2022) used RSMs and a learning procedure for learning a stabilizing policy together with an RSM that certifies a.s. asymptotic stability. However, as discussed in Section 1, this method is applicable only to systems in which the stabilizing region is assumed to be closed under system dynamics. In contrast, we propose the first method that does not require this assumption.

**Learning stable dynamics** Learning dynamics from observation data is the first step in model-based RL. Recent works considered learning deterministic Kolter & Manek (2019) and stochastic Umlauft & Hirche (2017); Lawrence et al. (2020) system dynamics with a specified stabilizing region.

**Safe exploration RL** Safe exploration RL restricts exploration of model-free RL algorithms in a way that ensures that given safety constraints are satisfied. This is typically ensured by learning the system dynamics' uncertainty and limiting exploratory actions within a high probability safe region via Gaussian Processes (Koller et al., 2018; Turchetta et al., 2019), linearized models Dalal et al. (2018), deep robust regression (Liu et al., 2020), and Bayesian neural networks (Lechner et al., 2021).

**Probabilistic program analysis** Ranking supermartingales were originally proposed for proving a.s. termination in probabilistic programs (PPs) (Chakarov & Sankaranarayanan, 2013). Since then, they have been used for termination (Chatterjee et al., 2016; Abate et al., 2021b) and safety (Chatterjee et al., 2017; Takisaka et al., 2021) analysis in PPs, and the work of (Chakarov et al., 2016) considers recurrence and persistence with the latter being equivalent to stability. However, the persistence certificate of (Chakarov et al., 2016) is numerically challenging for learning and it differs substantially from our notion of sRSMs.

## 3 PRELIMINARIES

We consider a discrete-time stochastic dynamical system of the form

$$\mathbf{x}_{t+1} = f(\mathbf{x}_t, \pi(\mathbf{x}_t), \omega_t),$$

where $f : \mathcal{X} \times \mathcal{U} \times \mathcal{N} \to \mathcal{X}$ is a dynamics function, $\pi : \mathcal{X} \to \mathcal{U}$ is a control policy and $\omega_t \in \mathcal{N}$ is a stochastic disturbance vector. Here, we use $\mathcal{X} \subseteq \mathbb{R}^n$ to denote the state space, $\mathcal{U} \subseteq \mathbb{R}^m$ the action space and $\mathcal{N} \subseteq \mathbb{R}^p$ the stochastic disturbance space of the system. In each time step, $\omega_t$ is sampled according to a probability distribution $d$ over $\mathcal{N}$, independently from the previous samples.

A sequence $(\mathbf{x}_t, \mathbf{u}_t, \omega_t)_{t \in \mathbb{N}_0}$ of state-action-disturbance triples is a trajectory of the system, if $\mathbf{u}_t = \pi(\mathbf{x}_t)$, $\omega_t \in \mathsf{support}(d)$ and $\mathbf{x}_{t+1} = f(\mathbf{x}_t, \mathbf{u}_t, \omega_t)$ hold for each $t \in \mathbb{N}_0$. For each state $\mathbf{x}_0 \in \mathcal{X}$, the system induces a Markov process and defines a probability space over the set of all trajectories that start in $\mathbf{x}_0$ Puterman (1994), with the probability measure and the expectation operators $\mathbb{P}_{\mathbf{x}_0}$ and $\mathbb{E}_{\mathbf{x}_0}$.

**Assumptions** The state space $\mathcal{X} \subseteq \mathbb{R}^n$, the action space $\mathcal{U} \subseteq \mathbb{R}^m$ and the stochastic disturbance space $\mathcal{N} \subseteq \mathbb{R}^p$ are all assumed to be Borel-measurable. Furthermore, we assume that the system has a *bounded maximal step size* under any policy $\pi$, i.e. that there exists $\Delta > 0$ such that for every $\mathbf{x} \in \mathcal{X}$, $\omega \in \mathcal{N}$ and policy $\pi$ we have $||\mathbf{x} - f(\mathbf{x}, \pi(\mathbf{x}), \omega)||_1 \leq \Delta$. Note that this is a realistic assumption that is satisfied in many real-world scenarios, e.g. a self-driving car can only traverse a certain maximal distance within each time step whose bounds depend on the maximal speed that the car can develop. For our learning procedure in Section 5, we also assume that $\mathcal{X} \subseteq \mathbb{R}^n$ is compact and that $f$ is Lipschitz continuous, which are common assumptions in control theory and RL.

**Almost-sure asymptotic stability** There are several notions of stability in stochastic systems. In this work, we consider the notion of almost-sure asymptotic stability (Kushner, 1965), which requires the system to eventually *converge and stay within* the stabilizing set. In order to define this formally, for each $\mathbf{x} \in \mathcal{X}$ let $d(\mathbf{x}, \mathcal{X}_s) = \inf_{\mathbf{x}_s \in \mathcal{X}_s} ||\mathbf{x} - \mathbf{x}_s||_1$, where $|| \cdot ||_1$ is the $l_1$-norm on $\mathbb{R}^m$.

**Definition 1.** *A non-empty Borel-measurable set $\mathcal{X}_s \subseteq \mathcal{X}$ is said to be* almost-surely (a.s.) asymptotically stable*, if for each initial state $\mathbf{x}_0 \in \mathcal{X}$ we have $\mathbb{P}_{\mathbf{x}_0}[\lim_{t \to \infty} d(\mathbf{x}_t, \mathcal{X}_s) = 0] = 1$.*

The above definition slightly differs from that of (Kushner, 1965) which considers the special case of $\mathcal{X}_s$ being a singleton set consisting only of the origin, i.e. $\mathcal{X}_s = \{\mathbf{0}\}$. The reason for this difference is that, analogously to (Lechner et al., 2022) and to the existing works on learning stabilizing policies in deterministic systems (Berkenkamp et al., 2017; Richards et al., 2018; Chang et al., 2019), we need to consider stability with respect to an open neighborhood of the origin for our learning method to be stable. Note that we *do not* assume that the stabilizing set $\mathcal{X}_s$ is closed under system dynamics so that the system cannot leave $\mathcal{X}_s$ once it is reached, which contrasts the previous works on stability in deterministic (Berkenkamp et al., 2017; Richards et al., 2018; Chang et al., 2019) and stochastic (Lechner et al., 2022) systems.

## 4 THEORETICAL RESULTS

In this section, we introduce our novel notion of stabilizing ranking supermartingales (sRSMs). We then show that sRSMs can be used to formally certify a.s. asymptotic stability with respect to a fixed policy *without* requiring that the stabilizing set is closed under system dynamics. Note, in this section only, we assume that the policy $\pi$ is fixed. In the next section, we will present our algorithm for learning policies that guarantee a.s. asymptotic stability together with an sRSM as a formal certificate of a.s. asymptotic stability.

**Overview of ranking supermartingales** In order to motivate our sRSMs and to explain their novelty, we first recall ranking supermartingales (RSMs) of (Lechner et al., 2022). RSMs were introduced for certifying a.s. asymptotic stability under a given policy $\pi$, when the stabilizing set is assumed to be closed under system dynamics. Note that, if the stabilizing set is assumed to be closed under system dynamics, then a.s. asymptotic stability of $\mathcal{X}_s$ is equivalent to *a.s. reachability* since the agent cannot leave $\mathcal{X}_s$ once entered. In what follows, we define RSMs and explain why they are insufficient for certifying a.s. asymptotic stability when the stabilizing set is not closed under system dynamics.

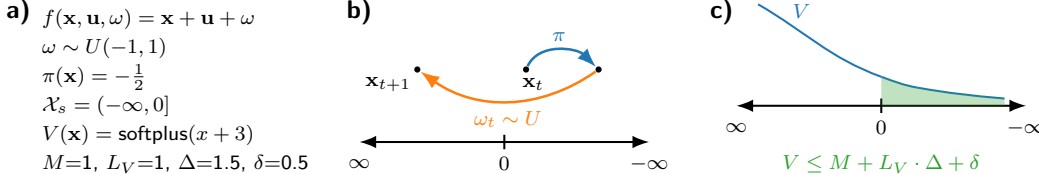

Figure 1: Example of a 1-dimensional stochastic dynamical system for which the stabilizing set $\mathcal{X}_s$ is not closed under system dynamics since from every system state any other state is reachable with positive probability. **a)** System definition and an sRSM that it admits. **b)** Illustration of a single time step evolution of the system. **c)** Visualization of the sRSM and the corresponding level set used to bound the probability of leaving the stabilizing region.

Intuitively, an RSM is a non-negative continuous function $V : \mathcal{X} \to \mathbb{R}$ that maps system states to non-negative real numbers and whose value at each state in $\mathcal{X} \backslash \mathcal{X}_s$ strictly decreases in expected value by some $\epsilon > 0$ upon every one-step evolution of the system under the policy $\pi$.

**Definition 2** (Ranking supermartingales (Lechner et al., 2022)). *A continuous function $V : \mathcal{X} \to \mathbb{R}$ is said to be a* ranking supermartingale (RSM) *for $\mathcal{X}_s$ if $V(\mathbf{x}) \geq 0$ holds for each $\mathbf{x} \in \mathcal{X}$ and if there exists $\epsilon > 0$ such that $\mathbb{E}_{\omega \sim d}[V(f(\mathbf{x}, \pi(\mathbf{x}), \omega))] \leq V(\mathbf{x}) - \epsilon$ holds for each $\mathbf{x} \in \mathcal{X} \backslash \mathcal{X}_s$.*

It was shown that, if a system under policy $\pi$ admits an RSM *and* the stabilizing set $\mathcal{X}_s$ is assumed to be closed under system dynamics, then $\mathcal{X}_s$ is a.s. asymptotically stable. The intuition behind this result is that $V$ needs to strictly decrease in expected value until $\mathcal{X}_s$ is reached while remaining bounded from below by 0. Results from martingale theory can then be used to prove that the agent must eventually converge and reach $\mathcal{X}_s$ with probability 1, due to a decrease in expected value by $\epsilon > 0$ outside of $\mathcal{X}_s$ being strict which prevents convergence to any other state. However, apart from nonnegativity, the defining conditions on RSMs do not impose any conditions on the RSM once the agent reaches $\mathcal{X}_s$. In particular, if the stabilizing set $\mathcal{X}_s$ is *not* closed under system dynamics, then the defining conditions of RSMs do not prevent the agent from leaving and reentering $\mathcal{X}_s$ infinitely many times and thus never stabilizing. In order to formally ensure stability, the defining conditions of RSMs need to be strengthened and in the rest of this section we solve this problem.

**Stabilizing ranking supermartingales** We now define our sRSMs, which may be used to certify a.s. asymptotic stability even when the stabilizing set is not assumed to be closed under system dynamics and thus overcome the limitation of RSMs of (Lechner et al., 2022) that was discussed above. Recall we use $\Delta$ to denote the maximal step size of the system.

**Definition 3** (Stabilizing ranking supermartingales). *Let $\epsilon, M, \delta > 0$. A Lipschitz continuous function $V : \mathcal{X} \to \mathbb{R}$ is said to be an $(\epsilon, M, \delta)$-stabilizing ranking supermartingale ($(\epsilon, M, \delta)$-sRSM) for $\mathcal{X}_s$ if the following three conditions hold:*

1. Nonnegativity. $V(\mathbf{x}) \geq 0$ *holds for each $\mathbf{x} \in \mathcal{X}$.*
2. Strict expected decrease if $V \geq M$. *For each $\mathbf{x} \in \mathcal{X}$, if $V(\mathbf{x}) \geq M$ then*

$$\mathbb{E}_{\omega \sim d}\Big[V\Big(f(\mathbf{x}, \pi(\mathbf{x}), \omega)\Big)\Big] \leq V(\mathbf{x}) - \epsilon.$$

3. Lower bound outside $\mathcal{X}_s$. $V(\mathbf{x}) \geq M + L_V \cdot \Delta + \delta$ *holds for each $\mathbf{x} \in \mathcal{X} \backslash \mathcal{X}_s$, where $L_V$ is a Lipschitz constant of $V$.*

An example of an sRSM for a 1-dimensional stochastic dynamical system is shown in Fig. 1. The intuition behind our new conditions is as follows. Condition 2 in Definition 3 requires that, at each state in which $V \geq M$, the value of $V$ decreases in expectation by $\epsilon > 0$ upon one-step evolution of the system. As we show below, this ensures probability 1 convergence to the set of states $S = \{\mathbf{x} \in \mathcal{X} \mid V(\mathbf{x}) \leq M\}$ from any other state of the system. On the other hand, condition 3 in Definition 3 requires that $V \geq M + L_V \cdot \Delta + \delta$ outside of the stabilizing set $\mathcal{X}_s$, thus $S \subseteq \mathcal{X}_s$. Moreover, if the agent is in a state where $V \leq M$, the value of $V$ in the next state has to be $\leq M + L_V \cdot \Delta$ due to Lipschitz continuity of $V$ and $\Delta$ being the maximal step size of the system. Therefore, even if the agent leaves $S$, for the agent to actually leave $\mathcal{X}_s$ the value of $V$ has to *increase* from a value $\leq M + L_V \cdot \Delta$ to a value $\geq M + L_V \cdot \Delta + \delta$ while satisfying the strict expected *decrease* condition imposed by condition 2 in Definition 3 at every intermediate state

that is not contained in $S$. The following theorem is the main result of this section and it shows that sRSMs indeed certify a.s. asymptotic stability of $\mathcal{X}_s$.

**Theorem 1.** *Suppose that there exist $\epsilon, M, \delta > 0$ and an $(\epsilon, M, \delta)$-sRSM for $\mathcal{X}_s$. Then $\mathcal{X}_s$ is a.s. asymptotically stable.*

The proof of the theorem and an overview of results from probability and martingale theory that we use in the proof are provided in Appendix A and B. In what follows, we outline the main ideas behind our proof. For each state $\mathbf{x}_0 \in \mathcal{X}$, we consider the probability space of all trajectories of the system that start in $\mathbf{x}_0$. We first show that the $(\epsilon, M, \delta)$-sRSM $V$ for $\mathcal{X}_s$ gives rise to an instance of the mathematical notion of supermartingales in this probability space. Next, we use Supermartingale Convergence Theorem (Williams, 1991) to show that Conditions 1 and 2 in Definition 3 ensure that the agent with probability 1 converges to the set of states $S = \{\mathbf{x} \in \mathcal{X} \mid V(\mathbf{x}) \leq M\} \subseteq \mathcal{X}_s$ from any other state in the system. Finally, we use a known concentration bound on the supremum value of a supermartingale process to show that the probability of the value of $V$ increasing from $\leq M + L_V \cdot \Delta$ to $\geq M + L_V \cdot \Delta + \delta$ is bounded from above by $p = \frac{M + L_V \cdot \Delta}{M + L_V \cdot \Delta + \delta}$. Hence, the agent with probability 1 converges to $S \subseteq \mathcal{X}_s$ from any state, upon which by Conditions 2 and 3 in Definition 3 it leaves $\mathcal{X}_s$ with probability at most $p < 1$. The probability of this happening $N$ times is at most $p^N$ so by letting $N \to \infty$ we conclude that the probability of the agent leaving $\mathcal{X}_s$ infinitely many times is 0. Therefore, the agent with probability 1 eventually stabilizes in $\mathcal{X}_s$.

**Bounds on stabilization time** We conclude this section by showing that our sRSMs not only certify a.s. asymptotic stability of $\mathcal{X}_s$, but also provide bounds on the number of time steps that the agent may spend outside of $\mathcal{X}_s$. This is particularly relevant for safety-critical applications in which the goal is not only to ensure stabilization but also to ensure that the agent spends as little time outside the stabilization set as possible. For each trajectory $\rho = (\mathbf{x}_t, \mathbf{u}_t, \omega_t)_{t \in \mathbb{N}_0}$, let $\mathsf{Out}_{\mathcal{X}_s}(\rho) = |\{t \in \mathbb{N}_0 \mid \mathbf{x}_t \notin \mathcal{X}_s\}| \in \mathbb{N}_0 \cup \{\infty\}$.

**Theorem 2.** *Let $\epsilon, M, \delta > 0$ and suppose that $V : \mathcal{X} \to \mathbb{R}$ is an $(\epsilon, M, \delta)$-sRSM for $\mathcal{X}_s$. Let $\Gamma = \sup_{\mathbf{x} \in \mathcal{X}_s} V(\mathbf{x})$ be the supremum of all possible values that $V$ can attain over the stabilizing set $\mathcal{X}_s$. Then, for each initial state $\mathbf{x}_0 \in \mathcal{X}$, we have that*

1. $\mathbb{E}_{\mathbf{x}_0}[\mathsf{Out}_{\mathcal{X}_s}] \leq \frac{V(\mathbf{x}_0)}{\epsilon} + \frac{(M + L_V \cdot \Delta) \cdot (\Gamma + L_V \cdot \Delta)}{\delta \cdot \epsilon}$.
2. $\mathbb{P}_{\mathbf{x}_0}[\mathsf{Out}_{\mathcal{X}_s} \geq t] \leq \frac{V(\mathbf{x}_0)}{t \cdot \epsilon} + \frac{(M + L_V \cdot \Delta) \cdot (\Gamma + L_V \cdot \Delta)}{\delta \cdot \epsilon \cdot t}$, *for any time $t \in \mathbb{N}$.*

*Proof.* See Appendix B.

## 5 Learning Stable Policies and sRSMs on Compact State Spaces

In this section, we present our method for learning a stabilizing policy together with an sRSM that certifies a.s. asymptotic stability. As stated in Section 3, our method assumes that the state space $\mathcal{X} \subseteq \mathbb{R}^n$ is compact and that $f$ is Lipschitz continuous with Lipschitz constant $L_f$.

We parameterize the policy and the sRSM via two neural networks $\pi_\theta : \mathcal{X} \to \mathcal{U}$ and $V_\nu : \mathcal{X} \to \mathbb{R}$. To enforce condition 1 in Definition 3, which requires the sRSM to be a nonnegative function, our method applies the softplus activation function $x \mapsto \log(\exp(x) + 1)$ to the output of $V_\nu$. The remaining layers of $\pi_\theta$ and $V_\nu$ apply ReLU activation functions, therefore $\pi_\theta$ and $V_\nu$ are also Lipschitz continuous (Szegedy et al., 2014). Our method draws insight from the algorithms of Chang et al. (2019); Žikelić et al. (2022) for learning policies together with Lyapunov functions or RSMs and it comprises of a *learner* and a *verifier* module that are composed into a loop. In each loop iteration, the learner module first trains both $\pi_\theta$ and $V_\nu$ on a training objective in the form of a differentiable approximation of the sRSM conditions 2 and 3 in Definition 3. Once the training has converged, the verifier module formally checks whether the learned sRSM candidate satisfies conditions 2 and 3 in Definition 3. If both conditions are fulfilled, our method terminates and returns a policy together with an sRSM witnessing stability. If at least one sRSM condition is violated, the verifier module enlarges the training set of the learner module by system states that violate the condition in order to guide the learner towards fixing the policy and the sRSM in the next learner iteration. The pseudocode of the algorithm is shown in Algorithm 1. In what follows, we provide details on initialization, the learner and the verifier modules.

---

**Algorithm 1** Procedure for learning a stabilizing policy and an sRSM

---

**Input** Dynamics function $f$, distribution $d$, stabilizing region $\mathcal{X}_s \subseteq \mathcal{X}$, Lipschitz constant $L_f$
**Parameters** $\tau > 0$, $N_{\text{cond 2}} \in \mathbb{N}$, $N_{\text{cond 3}} \in \mathbb{N}$, $M = 1$, $\epsilon_{\text{train}}$, $\delta_{\text{train}}$
$\pi_\theta \leftarrow$ policy trained by using PPO Schulman et al. (2017) on MDP $(\mathcal{X}, \mathcal{U}, f, x \mapsto \mathbb{1}[x \in \mathcal{X}_s])$
$\widetilde{\mathcal{X}} \leftarrow$ centers of grid cells of a discretization of $\mathcal{X}$ with mesh $\tau$
$B \leftarrow$ centers of grid cells of a subgrid of $\widetilde{\mathcal{X}}$
**while** timeout not reached **do**
    $\pi_\theta, V_\nu \leftarrow$ jointly trained by minimizing the loss function in eq. equation 1 on dataset $B$
    $L_\pi, L_V \leftarrow$ Lipschitz constants of $\pi_\theta, V_\nu$
    $K \leftarrow L_V \cdot (L_f \cdot (L_\pi + 1) + 1)$
    $\widetilde{\mathcal{X}}_{\geq M} \leftarrow$ centers of grid cells whose at least one vertex $\mathbf{x}$ satisfies $V_\nu(\mathbf{x}) \geq M$
    $\mathcal{X}_{ce} \leftarrow$ counterexamples to condition 2 in Definition 3 on $\widetilde{\mathcal{X}}_{\geq M}$
    **if** $\mathcal{X}_{ce} = \{\}$ **then**
        $\text{Cells}_{\mathcal{X} \setminus \mathcal{X}_s} \leftarrow$ grid cells that intersect $\mathcal{X} \setminus \mathcal{X}_s$
        $\Delta_\theta \leftarrow$ the maximal step size of the system with the policy $\pi$
        **if** $\underline{V}_\nu(\text{cell}) > M + L_V \cdot \Delta_\theta$ for all cell $\in \text{Cells}_{\mathcal{X} \setminus \mathcal{X}_s}$ **then**
            **return** $\mathcal{X}_s$ is a.s. asymptotically stable under policy $\pi_\theta$
        **end if**
    **else**
        $B \leftarrow (B \setminus \{\mathbf{x} \in B | V_\nu(\mathbf{x}) < M\}) \cup \mathcal{X}_{ce}$
    **end if**
**end while**
**Return** Unknown

---

**Initialization** We initialize the policy $\pi_\theta$ by running several iterations of the proximal policy optimization (PPO) Schulman et al. (2017) RL algorithm. In particular, we induce a Markov decision process (MDP) from the given system by using the reward function $x \mapsto \mathbb{1}[x \in \mathcal{X}_s]$ in order to learn an initial policy that drives the system toward the stabilizing set. The importance of initialization was observed in (Chang et al., 2019). As for the training set $B$ used by the learner, we discretize the state space $\mathcal{X}$ by using a rectangular grid and define $B$ to be the set of all centers of grid cells (discretization is defined formally below). Finally, note that we may always rescale an sRSM by a strictly positive constant factor. Therefore, without loss of generality, we assume the value $M = 1$ in Definition 3 for our sRSM.

**Learner** The policy and the sRSM candidate are learned by minimizing the loss

$$\mathcal{L}(\theta, \nu) = \mathcal{L}_{\text{cond 2}}(\theta, \nu) + \mathcal{L}_{\text{cond 3}}(\theta, \nu). \tag{1}$$

The two loss terms guide the learner toward an sRSM candidate that satisfies conditions 2 and 3 in Definition 3. In particular, we set

$$\mathcal{L}_{\text{cond 2}}(\theta, \nu) = \frac{1}{|B|} \sum_{\mathbf{x} \in B} \left( \max \left\{ \sum_{\omega_1, \ldots, \omega_{N_{\text{cond 2}}} \sim d} \frac{V_\nu\big(f(\mathbf{x}, \pi_\theta(\mathbf{x}), \omega_i)\big)}{N_{\text{cond 2}}} - V_\nu(\mathbf{x}) + \epsilon_{\text{train}}, 0 \right\} \right).$$

Intuitively, for each $\mathbf{x} \in B$, the corresponding term in the sum incurs a loss whenever condition 2 is violated at $\mathbf{x}$. Since the expected value of $V_\nu$ at a successor state of $\mathbf{x}$ does not admit a closed form expression due to $V_\nu$ being a neural network, we approximate it as the mean of values of $V_\nu$ at $N_{\text{cond 2}}$ independently sampled successor states of $\mathbf{x}$, with $N_{\text{cond 2}}$ being an algorithm parameter. For condition 3, the loss term samples $N_{\text{cond 3}}$ system states from $\mathcal{X} \setminus \mathcal{X}_s$ with $N_{\text{cond 3}}$ an algorithm parameter and incurs a loss whenever condition 3 is not satisfied at some sampled state:

$$\mathcal{L}_{\text{cond3}}(\theta, \nu) = \max\{(M + L_{V_\nu} + \Delta_\theta + \delta_{\text{train}}) - \min_{x_1, \ldots x_{N_{\text{cond 3}}} \sim \mathcal{X} \setminus \mathcal{X}_s} V_\nu(x_i), 0\}.$$

In our implementation, we also add two regularization terms to the loss function used by the learner. The first term favors learning an sRSM candidate whose global minimum is within the stabilizing set. The second term penalizes large Lipschitz bounds of the networks $\pi_\theta$ and $V_\nu$ by adding a regularization term. While these two loss terms do not directly enforce any particular condition in Definition 3, we observe that they help the learning and the verification process. Details on the regularization terms can be found in the Supplementary Material.

**Verifier** The verifier checks whether the learned sRSM candidate satisfies conditions 2 and 3 in Definition 3 (condition 1 is satisfied due to the softplus function applied to the outputs of $V_\nu$). The key challenge is checking the expected decrease condition imposed by condition 2. To check this condition, following the idea of Berkenkamp et al. (2017) and (Lechner et al., 2022) our method computes a *discretization* $\widetilde{\mathcal{X}}$ of $\mathcal{X}$ with *mesh* $\tau > 0$ so that for every $\mathbf{x} \in \mathcal{X}$ there exists $\widetilde{\mathbf{x}} \in \widetilde{\mathcal{X}}$ such that $||\widetilde{\mathbf{x}} - \mathbf{x}||_1 < \tau$. The discretization is computed by considering centers of cells of a rectangular grid of sufficiently small cell size. Then, due to the assumptions that the state space is compact and $f$, $\pi_\theta$ and $V_\nu$ are all Lipschitz continuous, we show that it suffices to verify a slightly stricter condition at discretization points.

To verify condition 2 in Definition 3, the verifier first collects the set $\widetilde{\mathcal{X}}_{\geq M}$ of centers of all grid cells whose at least one state $\mathbf{x}$ satisfies $V_\nu(\mathbf{x}) \geq M$. This set is computed via interval arithmetic abstract interpretation (IA-AI) (Cousot & Cousot, 1977; Gowal et al., 2018), which for each grid cell propagates interval bounds across neural network layers in order to bound from below the minimal value that $V_\nu$ attains over that cell and adds the center of a cell to $\widetilde{\mathcal{X}}_{\geq M}$ whenever this lower bound is smaller than $M$. Once $\widetilde{\mathcal{X}}_{\geq M}$ is computed, the verifier checks for each $\widetilde{\mathbf{x}} \in \widetilde{\mathcal{X}}_{\geq M}$ whether the following inequality holds

$$\mathbb{E}_{\omega \sim d}\Big[V_\nu\big(f(\widetilde{\mathbf{x}}, \pi_\theta(\widetilde{\mathbf{x}}), \omega)\big)\Big] < V_\nu(\widetilde{\mathbf{x}}) - \tau \cdot K, \tag{2}$$

where $L_\pi$ and $L_V$ are the Lipschitz constants of $\pi_\theta$ and $V_\nu$ and $K = L_V \cdot (L_f \cdot (L_\pi + 1) + 1)$. We use the method of (Szegedy et al., 2014) to compute $L_\pi$ and $L_V$. The reason behind checking this stronger constraint is that, due to Lipschitz continuity of all involved functions and due to $\tau$ being the mesh of the discretization, we can show (formally done in Theorem 3) that this condition being satisfied for each $\widetilde{\mathbf{x}} \in \widetilde{\mathcal{X}}_{\geq M}$ implies that the expected decrease condition $\mathbb{E}_{\omega \sim d}[V_\nu(f(\mathbf{x}, \pi_\theta(\widetilde{\mathbf{x}}), \omega))] < V_\nu(\mathbf{x})$ is satisfied for all $\mathbf{x} \in \mathcal{X}$ with $V(\mathbf{x}) \geq M$. Then, due to both sides of the inequality being continuous functions and $\{\mathbf{x} \in \mathcal{X} \mid V_\nu(\mathbf{x}) \geq M\}$ being a compact set, their difference admits a strictly positive global minimum $\epsilon > 0$ so that $\mathbb{E}_{\omega \sim d}[V_\nu(f(\mathbf{x}, \pi_\theta(\widetilde{\mathbf{x}}), \omega))] \leq V_\nu(\mathbf{x}) - \epsilon$ is satisfied for all $\mathbf{x} \in \mathcal{X}$ with $V(\mathbf{x}) \geq M$. If eq. equation 2 is satisfied for each $\widetilde{\mathbf{x}} \in \widetilde{\mathcal{X}}_{\geq M}$, the verifier concludes that $V_\nu$ satisfies condition 2 in Definition 3. Otherwise, any computed counterexample to this constraint is added to $B$ to help the learner fine-tune an sRSM candidate in the following learning iteration.

To formally check eq. equation 2 at some $\widetilde{\mathbf{x}} \in \widetilde{\mathcal{X}}_{\geq M}$, we need to compute an upper bound on the expected value $\mathbb{E}_{\omega \sim d}[V_\nu(f(\widetilde{\mathbf{x}}, \pi_\theta(\widetilde{\mathbf{x}}), \omega))]$. Note that this expected value does not admit a closed form expression due to $V_\nu$ being a neural network function. Thus, we again employ IA-AI to compute an upper bound on the expected value of a neural network function over a probability distribution. First, we partition the disturbance space $\mathcal{N} \subseteq \mathbb{R}^p$ into a grid of a finite amount of cells cell$(\mathcal{N}) = \{\mathcal{N}_1, \dots, \mathcal{N}_k\}$. We denote $\mathrm{maxvol} = \max_{\mathcal{N}_i \in \mathrm{cell}(\mathcal{N})} \mathrm{vol}(\mathcal{N}_i)$ the maximal volume of any cell in the partition. The expected value can then be bounded via

$$\mathbb{E}_{\omega \sim d}\Big[V_\nu\big(f(\widetilde{\mathbf{x}}, \pi_\theta(\widetilde{\mathbf{x}}), \omega)\big)\Big] \leq \sum_{\mathcal{N}_i \in \mathrm{cell}(\mathcal{N})} \mathrm{maxvol} \cdot \sup_{\omega \in \mathcal{N}_i} F(\omega)$$

where $F(\omega) = V_\nu(f(\widetilde{\mathbf{x}}, \pi_\theta(\widetilde{\mathbf{x}}), \omega)$ and the supremum values are obtained by using the IA-AI-based method of Gowal et al. (2018). Note that $\mathrm{maxvol}$ is infinite in case $\mathcal{N}$ is unbounded. To compute the expected value of an unbounded $\mathcal{N}$ when assuming that $d$ is a product of univariate distributions, we apply the probability integral transform Murphy (2012) to each univariate probability distribution in $d$. As a result, the problem is reduced to the case of a probability distribution of bounded support.

To verify condition 3 in Definition 3, the verifier collects the set $\mathrm{Cells}_{\mathcal{X} \setminus \mathcal{X}_s}$ of all grid cells that intersect $\mathcal{X} \setminus \mathcal{X}_s$. Then, for each $\mathrm{cell} \in \mathrm{Cells}_{\mathcal{X} \setminus \mathcal{X}_s}$, it uses IA-AI to check

$$\underline{V}_\nu(\mathrm{cell}) > M + L_V \cdot \Delta_\theta, \tag{3}$$

with $\underline{V}_\nu(\mathrm{cell})$ denoting the lower bound on $V_\nu$ over cell computed by IA-AI. If this holds, then the verifier concludes that $V_\nu$ satisfies condition 3 in Definition 3 with $\delta = \min_{\mathrm{cell} \in \mathrm{Cells}_{\mathcal{X} \setminus \mathcal{X}_s}} \{\underline{V}_\nu(\mathrm{cell}) - M - L_V \cdot \Delta_\theta\}$. Otherwise, it proceeds to the next learning iteration. The following theorem establishes the correctness of the verifier module.

**Theorem 3.** *Suppose that the verifier shows that $V_\nu$ satisfies eq. equation 2 for each $\widetilde{\mathbf{x}} \in \widetilde{\mathcal{X}}_{\geq M}$ and eq. equation 3 for each $\mathrm{cell} \in \mathrm{Cells}_{\mathcal{X} \setminus \mathcal{X}_s}$. Then $V_\nu$ is an sRSM and $\mathcal{X}_s$ is a.s. asymptotically stable under $\pi_\theta$.*

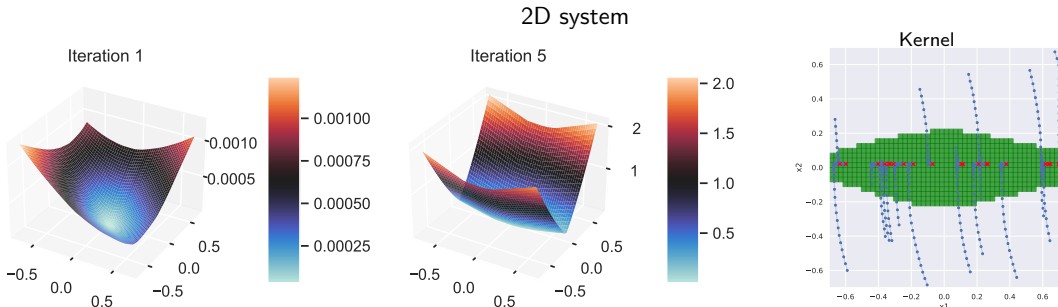

Figure 2: Visualization of the sRSM candidate after 1 and 5 iterations of our algorithm for the 2D system task. The candidate after 1 iteration does not fulfill all sRSM conditions, while the function after 5 learning iterations is a valid sRSM. The plot on the right shows the learned stabilizing subset (kernel) in green.

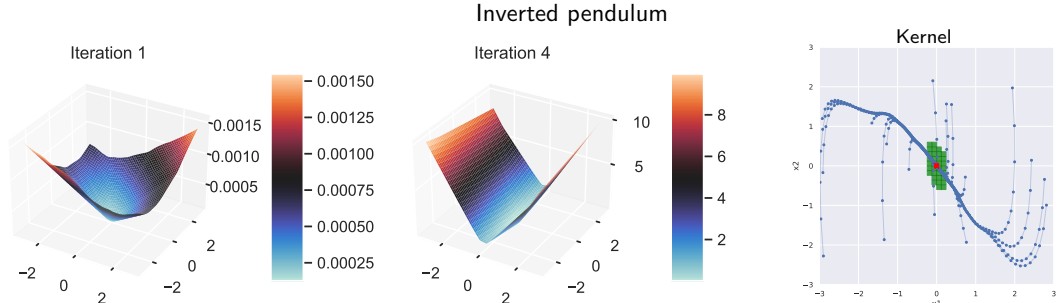

Figure 3: Visualization of the sRSM candidate after 1 and 4 iterations of our algorithm for the inverted pendulum task. The candidate after 1 iteration does not fulfill all sRSM conditions, while the function after 4 learning iterations is a valid sRSM. The plot on the right shows the learned stabilizing subset (kernel) in green.

*Proof.* See Appendix D.

## 6 EXPERIMENTAL RESULTS

In this section, we experimentally evaluate the effectiveness of our learning algorithm. We focus on the two benchmarks studied in Lechner et al. (2022). In particular, the authors could prove the stability of both systems when assuming that the stabilizing set is closed under system dynamics. However, both environments violate this assumption. Here, we aim to prove stability without assuming a given set that is closed under the system dynamics. We parameterize both $\pi_\theta$ and $V_\nu$ by two fully-connected networks with 2 hidden ReLU layers with 128 units each. The first task is a two-dimensional linear dynamical system with non-linear control bounds and is of the form $x_{t+1} = Ax_t + Bg(u_t) + \omega$, where $\omega$ is a disturbance vector sampled from a zero-mean triangular distribution. The function $g$ clips the action to stay within the interval [1, -1]. The state space is $\mathcal{X} = \{x \mid |x_1| \leq 0.7, |x_2| \leq 0.7\}$ and we want to learn a policy for the stabilizing set

$$\mathcal{X}_s = \mathcal{X}\backslash(\{x \mid -0.7 \leq x_1 \leq -0.6, -0.7 \leq x_2 \leq -0.4\} \cup \{x \mid 0.6 \leq x_1 \leq 0.7, 0.4 \leq x_2 \leq 0.7\}).$$

The second benchmark is a modified version of the inverted pendulum problem adapted from the OpenAI gym Brockman et al. (2016). The system is expressed by two state variables that represent the angle and the angu-

| Environment | Iterations | Mesh ($\tau$) | $p$ | Runtime |
|---|---|---|---|---|
| 2D system | 5 | 0.0007 | 0.80 | 3660 s |
| Inverted pendulum | 4 | 0.003 | 0.97 | 2619 s |

Table 1: Runtime statistics of our algorithm for both benchmarks.

lar velocity of the pendulum.

Contrary to the original task, the problem considered here introduces triangular-shaped random noise to the state after each update step.

The state space is define as $\mathcal{X} = \{x \mid |x_1| \leq 3, |x_2| \leq 3\}$, and objective of the agent is to stabilize within the set

$$\mathcal{X}_s = \mathcal{X} \backslash (\{x \mid -3 \leq x_1 \leq -2.9, -3 \leq x_2 \leq 0\} \cup \{x \mid 2.9 \leq x_1 \leq 3, 0 \leq x_2 \leq 3\}).$$

Further details for both tasks as well as additional plots are provided in the Supplementary Material.

For both tasks, our algorithm could find valid sRSMs and prove stability. The runtime characteristics, such as the number of iterations and total runtime, is shown in Table 4. In Figure 2 we plot the sRSM found by our algorithm for the 2D system task and in Figure 3 we plot the sRSM found for the inverted pendulum task. We also visualize in Figure 2 and Figure 3 in green the subset of $\mathcal{X}_s$ implied by the learned sRSM in which the system stabilizes for both of our example tasks. Finally, in Figure 4 we show the contour lines of the expected stabilization time bounds that are obtained by applying Theorem 2 to the learned sRSMs.

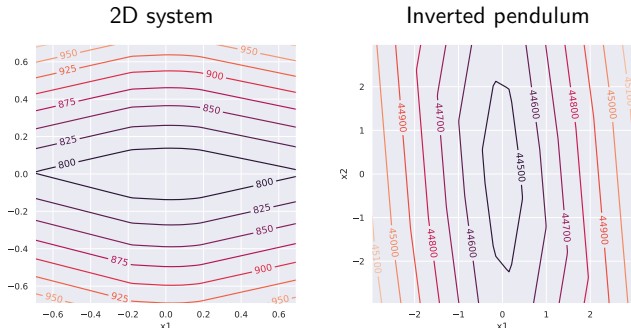

Figure 4: Contour lines of the expected stabilization time implied by Theorem 2 for the 2D system task on the left and the inverted pendulum task on the right.

**Limitations** Verification of neural networks is inherently a computationally difficult problem Katz et al. (2017); Berkenkamp et al. (2017); Sälzer & Lange (2021). Our method is subject to this barrier as well. In particular, the complexity of the grid decomposition routine for checking the expected decrease condition is exponential in the dimension of the system state space. However, a key advantage of our approach is that the complexity is only linear in the size of the neural network policy. Consequently, our approach allows learning and verifying networks that are of the size of typical networks used in reinforcement learning Schulman et al. (2017). Moreover, our grid decomposition procedure runs entirely on accelerator devices, including CPUs, GPUs, and TPUs, thus leveraging future advances in these computing devices. A technical limitation of our learning procedure is that it is restricted to compact state spaces. However, this is a standard assumption in control theory and reinforcement learning. Our theoretical results are applicable to arbitrary (potentially unbounded) state spaces, as shown in Fig. 1.

## 7 CONCLUSION

In this work, we developed a method for learning policies for stochastic control systems with formal guarantees about the systems' a.s. asymptotic stability over the infinite time horizon. Compared to the existing literature, which assumes that the stabilizing set is closed under system dynamics and cannot be left once entered, our approach does not impose this assumption. Our method is based on the novel notion of stabilizing ranking supermartingales (sRSMs) that serve as a formal certificate of a.s. asymptotic stability. We experimentally showed that our learning procedure is able to learn stabilizing policies and stability proof certificates in practice.

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

## A  OVERVIEW OF PROBABILITY AND MARTINGALE THEORY

**Probability theory** A *probability space* is an ordered triple $(\Omega, \mathcal{F}, \mathbb{P})$ consisting of a non-empty *sample space* $\Omega$, a *$\sigma$-algebra* $\mathcal{F}$ over $\Omega$ (i.e. a collection of subsets of $\Omega$ that contains the empty set $\emptyset$ and is closed under complementation and countable union), and a *probability measure* $\mathbb{P}$ over $\mathcal{F}$ which is a function $\mathbb{P} : \mathcal{F} \rightarrow [0,1]$ that satisfies the three Kolmogorov axioms: (1) $\mathbb{P}[\emptyset] = 0$, (2) $\mathbb{P}[\Omega \backslash A] = 1 - \mathbb{P}[A]$ for each $A \in \mathcal{F}$, and (3) $\mathbb{P}[\cup_{i=0}^{\infty} A_i] = \sum_{i=0}^{\infty} \mathbb{P}[A_i]$ for any sequence $(A_i)_{i=0}^{\infty}$ of pairwise disjoint sets in $\mathcal{F}$. Given a probability space $(\Omega, \mathcal{F}, \mathbb{P})$, a *random variable* is a function $X : \Omega \rightarrow \mathbb{R} \cup \{\pm\infty\}$ that is $\mathcal{F}$-measurable, i.e. for each $a \in \mathbb{R}$ we have $\{\omega \in \Omega \mid X(\omega) \leq a\} \in \mathcal{F}$. $\mathbb{E}[X]$ denotes the *expected value* of $X$. A *(discrete-time) stochastic process* is a sequence $(X_i)_{i=0}^{\infty}$ of random variables in $(\Omega, \mathcal{F}, \mathbb{P})$.

**Conditional expectation** Let $(\Omega, \mathcal{F}, \mathbb{P})$ be a probability space and $X$ be a random variable in $(\Omega, \mathcal{F}, \mathbb{P})$. Given a sub-sigma-algebra $\mathcal{F}' \subseteq \mathcal{F}$, a *conditional expectation* of $X$ given $\mathcal{F}'$ is an $\mathcal{F}'$-measurable random variable $Y$ such that, for each $A \in \mathcal{F}'$, we have

$$\mathbb{E}[X \cdot \mathbb{I}_A] = \mathbb{E}[Y \cdot \mathbb{I}_A].$$

Here $\mathbb{I}_A : \Omega \rightarrow \{0, 1\}$ is an *indicator function* of $A$, defined via $\mathbb{I}_A(\omega) = 1$ if $\omega \in A$, and $\mathbb{I}_A(\omega) = 0$ if $\omega \notin A$. If $X$ is real-valued and nonnegative, then a conditional expectation of $X$ given $\mathcal{F}'$ exists and is almost-surely unique, i.e. for any two $\mathcal{F}'$-measurable random variables $Y$ and $Y'$ which are conditional expectations of $X$ given $\mathcal{F}'$ we have that $\mathbb{P}[Y = Y'] = 1$ (Williams, 1991). Therefore, we may pick any such random variable as a canonical conditional expectation and denote it by $\mathbb{E}[X \mid \mathcal{F}']$.

**Stopping time** A sequence of sigma-algebras $\{\mathcal{F}_i\}_{i=0}^{\infty}$ with $\mathcal{F}_0 \subseteq \mathcal{F}_1 \subseteq \cdots \subseteq \mathcal{F}$ is a *filtration* in the probability space $(\Omega, \mathcal{F}, \mathbb{P})$. A *stopping time* with respect to a filtration $\{\mathcal{F}_i\}_{i=0}^{\infty}$ is a random variable $T : \Omega \rightarrow \mathbb{N}_0 \cup \{\infty\}$ such that, for every $i \in \mathbb{N}_0$, we have $\{\omega \in \Omega \mid T(\omega) \leq i\} \in \mathcal{F}_i$. Intuitively, $T$ may be viewed as the time step at which some stochastic process should be "stopped", and since $\{\omega \in \Omega \mid T(\omega) \leq i\} \in \mathcal{F}_i$ the decision to stop at the time step $i$ is made solely by using the information available in the first $i$ time steps.

**Supermartingales and ranking supermartingales** We now define the mathematical notion of ranking supermartingales. Let $(\Omega, \mathcal{F}, \mathbb{P})$ be a probability space, let $\epsilon \geq 0$ and let $T$ be a stopping time with respect to a filtration $\{\mathcal{F}_i\}_{i=0}^{\infty}$. An *$\epsilon$-ranking supermartingale ($\epsilon$-RSM) with respect to $T$* is a stochastic process $(X_i)_{i=0}^{\infty}$ such that

- $X_i$ is $\mathcal{F}_i$-measurable, for each $i \geq 0$,
- $X_i(\omega) \geq 0$, for each $i \geq 0$ and $\omega \in \Omega$, and
- $\mathbb{E}[X_{i+1} \mid \mathcal{F}_i](\omega) \leq X_i(\omega) - \epsilon \cdot \mathbb{I}_{T>i}(\omega)$, for each $i \geq 0$ and $\omega \in \Omega$.

The name comes since RSMs are a special instance of classical *supermartingale processes* Williams (1991). A *supermartingale* with respect to a filtration $\{\mathcal{F}_i\}_{i=0}^{\infty}$ is a stochastic process $(X_i)_{i=0}^{\infty}$ which satisfies conditions 1 and 3 above with $\epsilon = 0$ (thus we define supermartingales only with respect to the filtration and not the stopping time). We conclude this overview with two results on RSMs and supermartingales that will later be used in our proofs. The first is a result on ranking supermartingales that was originally presented in works on termination analysis of probabilistic programs Fioriti & Hermanns (2015); Chatterjee et al. (2016). The second result (see Kushner (2014), Theorem 7.1) is a concentration bound on the supremum value of a nonnegative supemartingale.

**Proposition 1.** *Let $(\Omega, \mathcal{F}, \mathbb{P})$ be a probability space, let $(\mathcal{F}_i)_{i=0}^{\infty}$ be a filtration and let $T$ be a stopping time with respect to $(\mathcal{F}_i)_{i=0}^{\infty}$. Suppose that $(X_i)_{i=0}^{\infty}$ is an $\epsilon$-RSM with respect to $T$, for some $\epsilon > 0$. Then*

1. $\mathbb{P}[T < \infty] = 1$,

2. $\mathbb{E}[T] \leq \frac{\mathbb{E}[X_0]}{\epsilon}$, *and*

3. $\mathbb{P}[T \geq t] \leq \frac{\mathbb{E}[X_0]}{\epsilon \cdot t}$, *for each $t \in \mathbb{N}$.*

**Proposition 2.** *Let $(\Omega, \mathcal{F}, \mathbb{P})$ be a probability space and let $(\mathcal{F}_i)_{i=0}^{\infty}$ be a filtration. Let $(X_i)_{i=0}^{\infty}$ be a nonnegative supermartingale with respect to $(\mathcal{F}_i)_{i=0}^{\infty}$. Then, for every $\lambda > 0$, we have*

$$\mathbb{P}\left[\sup_{i \geq 0} X_i \geq \lambda\right] \leq \frac{\mathbb{E}[X_0]}{\lambda}.$$

## B  PROOFS OF THEOREM 1 AND THEOREM 2

We now prove Theorem 1 and Theorem 2 from the main text of the paper. For each initial state $\mathbf{x}_0 \in \mathcal{X}$, denote by $(\Omega_{\mathbf{x}_0}, \mathcal{F}_{\mathbf{x}_0}, \mathbb{P}_{\mathbf{x}_0})$ probability space over the set of all system trajectories that start in the initial state $\mathbf{x}_0$ that is induced by the Markov decision process semantics of the system (Puterman, 1994). The key idea behind both proofs is to show that, for every state $\mathbf{x}_0 \in \mathcal{X} \backslash \mathcal{X}_s$, the sRSM $V$ for the set $\mathcal{X}_s$ gives rise to a mathematical RSM in the probability space $(\Omega_{\mathbf{x}_0}, \mathcal{F}_{\mathbf{x}_0}, \mathbb{P}_{\mathbf{x}_0})$. We then use Proposition 1 and Proposition 2 to prove the claims of both theorems.

**Canonical filtration and stopping time** In order to formally show that $V$ can be instantiated as a mathematical RSM in this probability space, we first define the canonical filtration in this probability space and the stopping time with respect to which the mathematical RSM is defined.

Let $\mathbf{x}_0 \in \mathcal{X}$ and consider the probability space $(\Omega_{\mathbf{x}_0}, \mathcal{F}_{\mathbf{x}_0}, \mathbb{P}_{\mathbf{x}_0})$. For each $i \in \mathbb{N}_0$, define $\mathcal{F}_i \subseteq \mathcal{F}$ to be the $\sigma$-algebra containing the subsets of $\Omega_{\mathbf{x}_0}$ that, intuitively, contain all trajectories in $\Omega_{\mathbf{x}_0}$ whose first $i$ states satisfy some specified property. Formally, we define $\mathcal{F}_i$ as follows. For each $j \in \mathbb{N}_0$, let $C_j : \Omega_{\mathbf{x}_0} \to \mathcal{X}$ be a map which to each trajectory $\rho = (\mathbf{x}_t, \mathbf{u}_t, \omega_t)_{t \in \mathbb{N}_0} \in \Omega_{\mathbf{x}_0}$ assigns the $j$-th state $\mathbf{x}_j$ along the trajectory. Then $\mathcal{F}_i$ is the smallest $\sigma$-algebra over $\Omega_{\mathbf{x}_0}$ with respect to which $C_0, C_1, \ldots, C_i$ are all measurable, where $\mathcal{X} \subseteq \mathbb{R}^m$ is equipped with the induced Borel-$\sigma$-algebra (Williams, 1991, Section 1). Clearly $\mathcal{F}_0 \subseteq \mathcal{F}_1 \subseteq \ldots$. We say that the sequence of $\sigma$-algebras $(\mathcal{F}_i)_{i=0}^{\infty}$ is the *canonical filtration* in the probability space $(\Omega_{\mathbf{x}_0}, \mathcal{F}_{\mathbf{x}_0}, \mathbb{P}_{\mathbf{x}_0})$.

We then define $T_S : \Omega_{\mathbf{x}_0} \to \mathbb{N}_0 \cup \{\infty\}$ to be the first hitting time of the set $S = \{\mathbf{x} \in \mathcal{X} \mid V(\mathbf{x}) \leq M\}$, i.e. $T_S = \inf\{t \in \mathbb{N}_0 \mid \mathbf{x}_t \in S\}$. Since whether $T_S(\rho) \leq i$ depends solely on the first $i$ states along $\rho$, we clearly have $\{\rho \in \Omega_{\mathbf{x}_0} \mid T_S(\rho) \leq i\} \in \mathcal{F}_i$ for each $i$ and so $T_S$ is a stopping time with respect to $(\mathcal{F}_i)_{i=0}^{\infty}$.

We now prove the theorems.

**Theorem.** *Suppose that there exist $\epsilon, M, \delta > 0$ and an $(\epsilon, M, \delta)$-sRSM for $\mathcal{X}_s$. Then $\mathcal{X}_s$ is a.s. asymptotically stable.*

*Proof.* In order to prove that $\mathcal{X}_s$ is a.s. asymptotically stable we need to show that, for each $\mathbf{x}_0 \in \mathcal{X}$,

$$\mathbb{P}_{\mathbf{x}_0}\left[\lim_{t \to \infty} d(\mathbf{x}_t, \mathcal{X}_s) = 0\right] = 1.$$

We prove the theorem statement by proving the following two claims. First, we show that, from each initial state $\mathbf{x}_0 \in \mathcal{X}$, the agent with probability 1 converges to and reaches $S = \{\mathbf{x} \in \mathcal{X} \mid V(\mathbf{x}) \leq M\}$ which is a subset of $\mathcal{X}_s$ by condition 3 in Definition 3 of sRSMs. Second, we show that once the agent is in $S$ it may leave $\mathcal{X}_s$ with probability at most $p = \frac{M + L_V \cdot \Delta}{M + L_V \cdot \Delta + \delta} < 1$. We then prove that the two claims imply the theorem statement.

*Claim 1.* For each $\mathbf{x}_0 \in \mathcal{X}$, $\mathbb{P}_{\mathbf{x}_0}[\exists t \in \mathbb{N}_0 \text{ s.t. } \mathbf{x}_t \in S] = 1$.

To prove Claim 1, let $\mathbf{x}_0 \in \mathcal{X}$. If $\mathbf{x}_0 \in S$, then the claim trivially holds. Thus suppose without loss of generality that $\mathbf{x}_0 \notin S$ so $V(\mathbf{x}_0) > M$, and consider the probability space $(\Omega_{\mathbf{x}_0}, \mathcal{F}_{\mathbf{x}_0}, \mathbb{P}_{\mathbf{x}_0})$, the canonical filtration $(\mathcal{F}_i)_{i=0}^{\infty}$ and the stopping time $T_S$ with respect to it.

Define a stochastic process $(X_i)_{i=0}^{\infty}$ in $(\Omega_{\mathbf{x}_0}, \mathcal{F}_{\mathbf{x}_0}, \mathbb{P}_{\mathbf{x}_0})$ via

$$X_i(\rho) = \begin{cases} V(\mathbf{x}_i), & \text{if } i < T_S(\rho) \\ V(\mathbf{x}_{T_S(\rho)}), & \text{otherwise} \end{cases}$$

for each $i \geq 0$ and $\rho = (\mathbf{x}_t, \mathbf{u}_t, \omega_t)_{t \in \mathbb{N}_0} \in \Omega_{\mathbf{x}_0}$. In other words, $X_i$ is equal to the value of $V$ at the $i$-th state along the trajectory until the stopping time $T_S$ is exceeded, after which $X_i$ is equal to the value of $V$ at the time step $T_S$ at which the process was stopped.

We prove that $(X_i)_{i=0}^{\infty}$ is an $\epsilon$-RSM with respect to the stopping time $T_S$. To prove this claim, we check each defining property of $\epsilon$-RSMs:

- *Each $X_i$ is $\mathcal{F}_i$-measurable.* The value of $X_i$ is determined by the first $i$ states along a trajectory, so by the definition of the canonical filtration we have that $X_i$ is $\mathcal{F}_i$-measurable for each $i \geq 0$.

- *Each $X_i(\rho) \geq 0$.* Since each $X_i$ is defined in terms of $V$ and since we know that $V(\mathbf{x}) \geq 0$ for each state $\mathbf{x} \in \mathcal{X}$ by condition 1 in Definition 3 of sRSMs, it follows that $X_i(\rho) \geq 0$ for each $i \geq 0$ and $\rho \in \Omega_{\mathbf{x}_0}$.

- *Each $\mathbb{E}[X_{i+1} \mid \mathcal{F}_i](\rho) \leq X_i(\rho) - \epsilon \cdot \mathbb{I}_{T_{\mathcal{X}_s} > i}(\rho)$.* First, we remark that the conditional expectation exists since $X_{i+1}$ is nonnegative for each $i \geq 0$. In order to prove the desired inequality, we distinguish between two cases. Let $\rho = (\mathbf{x}_t, \mathbf{u}_t, \omega_t)_{t \in \mathbb{N}_0}$.

  First, consider the case $T_S(\rho) > i$. We have that $X_i(\rho) = V(\mathbf{x}_i)$. On the other hand, we have $\mathbb{E}[X_{i+1} \mid \mathcal{F}_i](\rho) = \mathbb{E}_{\omega \sim d}[V(f(\mathbf{x}_i, \pi(\mathbf{x}_i), \omega)]$. To see this, observe that $\mathbb{E}_{\omega \sim d}[V(f(\mathbf{x}_i, \pi(\mathbf{x}_i), \omega)]$ satisfies all the defining properties of conditional expectation since it is the expected value of $V$ at a subsequent state of $\mathbf{x}_i$, and recall that conditional expectation is a.s. unique whenever it exists. Hence,

$$\mathbb{E}[X_{i+1} \mid \mathcal{F}_i](\rho) = \mathbb{E}_{\omega \sim d}[V(f(\mathbf{x}_i, \pi(\mathbf{x}_i), \omega)]$$
$$\leq V(\mathbf{x}_i) - \epsilon = X_i(\rho) - \epsilon,$$

  where the inequality holds by condition 2 in Definition 3 of sRSMs and since $\mathbf{x}_i \notin S$ as $T_S(\rho) > i$. This proves the desired inequality.

  Second, consider the case $T_S(\rho) \leq i$. We have $X_i(\rho) = V(\mathbf{x}_{T_S(\rho)})$ and $\mathbb{E}[X_{i+1} \mid \mathcal{F}_i](\rho)] = V(\mathbf{x}_{T_S(\rho)})$, so the desired inequality follows.

Thus, we may use the first part of Proposition 1 to conclude that $\mathbb{P}_{\mathbf{x}_0}[T_S < \infty] = 1$, equivalently $\mathbb{P}_{\mathbf{x}_0}[\exists t \in \mathbb{N}_0 \text{ s.t. } \mathbf{x}_t \in S] = 1$. This concludes the proof of Claim 1.

*Claim 2.* For each $\mathbf{x}_0 \in S$, $\mathbb{P}_{\mathbf{x}_0}[\exists t \in \mathbb{N}_0 \text{ s.t. } \mathbf{x}_t \notin \mathcal{X}_s] = p < 1$ with $p = \frac{M + L_V \cdot \Delta}{M + L_V \cdot \Delta + \delta}$.

To prove Claim 2, recall that $S = \{\mathbf{x} \in \mathcal{X} \mid V(\mathbf{x}) \leq M\}$. Thus, as $V$ is Lipschitz continuous with Lipschitz constant $L_V$ and as $\Delta$ is the maxmial step size of the system, it follows that the value of $V$ upon the agent leaving the set $S$ is $\leq M + L_V \cdot \Delta$. Hence, for the agent to leave $\mathcal{X}_s$ from $\mathbf{x}_0 \in S$, it first has to reach a state $\mathbf{x}_1$ with $M < V(\mathbf{x}_1) \leq M + L_V \cdot \Delta$ and then also to reach a state $\mathbf{x}_2 \notin \mathcal{X}_s$ from $\mathbf{x}_1$ without reentering $S$. By condition 3 in Definition 3 of sRSMs, we must

have $V(\mathbf{x}_2) \geq M + L_V \cdot \Delta + \delta$. Therefore,

$$\mathbb{P}_{\mathbf{x}_0}\Big[\exists t \in \mathbb{N}_0 \text{ s.t. } \mathbf{x}_t \notin \mathcal{X}_s\Big]$$

$$=\mathbb{P}_{\mathbf{x}_0}\Big[\exists t_1, t_2 \in \mathbb{N}_0 \text{ s.t. } t_1 < t_2 \text{ and } M < V(\mathbf{x}_{t_1}) \leq M + L_V \cdot \Delta \text{ and } V(\mathbf{x}_2) \geq M + L_V \cdot \Delta + \delta$$

$$\text{with } \mathbf{x}_t \notin S \text{ for all } t_1 \leq t \leq t_2\Big]$$

$$=\mathbb{P}_{\mathbf{x}_0}\Big[\exists t_1 \in \mathbb{N}_0 \text{ s.t. } M < V(\mathbf{x}_{t_1}) \leq M + L_V \cdot \Delta\Big]$$

$$\cdot \mathbb{P}_{\mathbf{x}_0}\Big[\exists t_1, t_2 \in \mathbb{N}_0 \text{ s.t. } t_1 < t_2 \text{ and } M < V(\mathbf{x}_{t_1}) \leq M + L_V \cdot \Delta \text{ and } V(\mathbf{x}_2) \geq M + L_V \cdot \Delta + \delta$$

$$\text{with } \mathbf{x}_t \notin S \text{ for all } t_1 \leq t \leq t_2 \mid \exists t_1 \in \mathbb{N}_0 \text{ s.t. } M < V(\mathbf{x}_{t_1}) \leq M + L_V \cdot \Delta\Big]$$

$$\leq\mathbb{P}_{\mathbf{x}_0}\Big[\exists t_1 \in \mathbb{N}_0 \text{ s.t. } M < V(\mathbf{x}_{t_1}) \leq M + L_V \cdot \Delta\Big]$$

$$\cdot \sup_{\mathbf{x}_1 \in \mathcal{X},\, M < V(\mathbf{x}_{t_1}) \leq M + L_V \cdot \Delta} \mathbb{P}_{\mathbf{x}_1}\Big[\exists t_2 \in \mathbb{N}_0 \text{ s.t. } V(\mathbf{x}_{t_2}) \geq M + L_V \cdot \Delta + \delta \text{ and } \mathbf{x}_t \notin S \text{ for all } 0 \leq t \leq t_2\Big]$$

$$\leq \sup_{\mathbf{x}_1 \in \mathcal{X},\, M < V(\mathbf{x}_{t_1}) \leq M + L_V \cdot \Delta} \mathbb{P}_{\mathbf{x}_1}\Big[\exists t_2 \in \mathbb{N}_0 \text{ s.t. } V(\mathbf{x}_{t_2}) \geq M + L_V \cdot \Delta + \delta \text{ and } \mathbf{x}_t \notin S \text{ for all } 0 \leq t \leq t_2\Big].$$

The first equality follows by the above observations. The second equality follows by Bayes' rule. The third inequality follows by observing that the trajectory satisfies the Markov property and therefore that the supremum value of $V$ upon visiting a state does not depend on previously visited states. Finally, the fourth inequality follows since the value of the first probability term is $\leq 1$.

Thus, to prove that $\mathbb{P}_{\mathbf{x}_0}[\exists t \in \mathbb{N}_0 \text{ s.t. } \mathbf{x}_t \notin \mathcal{X}_s] = p < 1$ with $p = \frac{M + L_V \cdot \Delta}{M + L_V \cdot \Delta + \delta}$ and therefore conclude Claim 2, it suffices to prove that, for each $\mathbf{x}_1 \in \mathcal{X}$ with $M < V(\mathbf{x}_{t_1}) \leq M + L_V \cdot \Delta$, we have

$$\mathbb{P}_{\mathbf{x}_1}\Big[\exists t_2 \in \mathbb{N}_0 \text{ s.t. } V(\mathbf{x}_{t_2}) \geq M + L_V \cdot \Delta + \delta \text{ and } \mathbf{x}_t \notin S \text{ for all } 0 \leq t \leq t_2\Big] \leq \frac{M + L_V \cdot \Delta}{M + L_V \cdot \Delta + \delta}.$$

To prove Claim 1, consider the probability space $(\Omega_{\mathbf{x}_1}, \mathcal{F}_{\mathbf{x}_1}, \mathbb{P}_{\mathbf{x}_1})$, the canonical filtration $(\mathcal{F}_i)_{i=0}^\infty$ and the stopping time $T_S$ with respect to it, and define a stochastic process $(X_i)_{i=0}^\infty$ in the probability space via

$$X_i(\rho) = \begin{cases} V(\mathbf{x}_i), & \text{if } i < T_S(\rho) \\ V(\mathbf{x}_{T_S(\rho)}), & \text{otherwise} \end{cases}$$

for each $i \geq 0$ and a trajectory $\rho$ that starts in $\mathbf{x}_1$. The argument analogous to the proof of Claim 1 shows that it is an $\epsilon$-RSM with respect to the stopping time $T_S$. But note that $\sup_{i \geq 0} X_i$ is equal to the supremum value attained by $V$ until the first hitting time of the set $S$. Hence the above inequality follows immediately from Proposition 2 by observing that $\mathbb{E}_{\mathbf{x}_1}[X_0] = V(\mathbf{x}_1) \leq M + L_V \cdot \Delta$ and plugging in $\lambda = M + L_V \cdot \Delta + \delta$. This concludes the proof of Claim 2.

*Proof that Claim 1 and Claim 2 imply Theorem 1.* By Claim 1, the agent with probability 1 converges to $S \subseteq \mathcal{X}_s$ from any initial state. On the other hand, by Claim 2, upon reaching a state in $S$ the probability of leaving $\mathcal{X}_s$ is at most $p < 1$. Finally, by Claim 1 again the agent is guaranteed to converge back to $S$ even upon leaving $\mathcal{X}_s$. Hence, due to the system dynamics under a given policy satisfying Markov property, the probability of the agent leaving and reentering $S$ more than $N$ times is bounded from above by $p^N$. Hence, by letting $N \to \infty$, we conclude that the probability of the agent leaving $\mathcal{X}_s$ and reentering infinitely many times is 0, so the agent with probability 1 eventually enters and $S$ and does not leave $\mathcal{X}_s$ after that. This implies that $\mathcal{X}_s$ is a.s. asymptotically stable.

$\square$

**Theorem.** *Let $\epsilon, M, \delta > 0$ and suppose that $V : \mathcal{X} \to \mathbb{R}$ is an $(\epsilon, M, \delta)$-sRSM for $\mathcal{X}_s$. Let $\Gamma = \sup_{\mathbf{x} \in \mathcal{X}_s} V(\mathbf{x})$ be the supremum of all possible values that $V$ can attain over the stabilizing set $\mathcal{X}_s$. Then, for each initial state $\mathbf{x}_0 \in \mathcal{X}$, we have that*

1. $\mathbb{E}_{\mathbf{x}_0}[\mathsf{Out}_{\mathcal{X}_s}] \leq \frac{V(\mathbf{x}_0)}{\epsilon} + \frac{(M + L_V \cdot \Delta) \cdot (\Gamma + L_V \cdot \Delta)}{\delta \cdot \epsilon}.$

2. $\mathbb{P}_{\mathbf{x}_0}[\mathsf{Out}_{\mathcal{X}_s} \geq t] \leq \frac{V(\mathbf{x}_0)}{t \cdot \epsilon} + \frac{(M + L_V \cdot \Delta) \cdot (\Gamma + L_V \cdot \Delta)}{\delta \cdot \epsilon \cdot t}$, *for any time* $t \in \mathbb{N}$.

*Proof.* We start by proving the first item in Theorem 2. Let $\rho = (\mathbf{x}_t, \mathbf{u}_t, \omega_t)_{t \in \mathbb{N}_0}$ be a system trajectory. Recall that $S = \{\mathbf{x} \in \mathcal{X} \mid V(\mathbf{x}) \leq M\} \subseteq \mathcal{X}_s$ and that $T_S(\rho) = \inf\{t \in \mathbb{N}_0 \mid \mathbf{x}_t \in \mathcal{X}_s\}$ is the first hitting time of $S$. Let us also denote by $\mathsf{OutAfter}_{\mathcal{X}_s}(\rho) = |\{t > T_S(\rho) \mid \mathbf{x}_t \notin \mathcal{X}_s\}|$ the number of time-steps that the trajectory $\rho$ is in states outside of the stabilizing set $\mathcal{X}_s$ after the first hitting time of $S$. Then, since $S \subseteq \mathcal{X}_s$, for each system trajectory $\rho = (\mathbf{x}_t, \mathbf{u}_t, \omega_t)_{t \in \mathbb{N}_0}$ we have that

$$\mathsf{Out}_{\mathcal{X}_s}(\rho) \leq T_S(\rho) + \mathsf{OutAfter}_{\mathcal{X}_s}(\rho).$$

Therefore, for each initial state $\mathbf{x}_0 \in \mathcal{X}$, we have

$$\begin{aligned} \mathbb{E}_{\mathbf{x}_0}[\mathsf{Out}_{\mathcal{X}_s}] &\leq \mathbb{E}_{\mathbf{x}_0}[T_S] + \mathbb{E}_{\mathbf{x}_0}[\mathsf{OutAfter}_{\mathcal{X}_s}] \\ &\leq \mathbb{E}_{\mathbf{x}_0}[T_S] + \sup_{\mathbf{x} \in \mathcal{X}} \mathbb{E}_{\mathbf{x}}[\mathsf{OutAfter}_{\mathcal{X}_s}]. \end{aligned} \tag{4}$$

Now, by defining an $\epsilon$-RSM $(X_i)_{i=0}^{\infty}$ with respect to the stopping time $T_S$ analogously as in the proof of Theorem 1 and by applying the second item in Proposition 1 to it, we can immediately deduce that

$$\mathbb{E}_{\mathbf{x}_0}[T_S] \leq \frac{\mathbb{E}_{\mathbf{x}_0}[X_0]}{\epsilon} = \frac{V(\mathbf{x}_0)}{\epsilon}. \tag{5}$$

On the other hand, by Claim 2 in the proof of Theorem 1 we know that the probability of leaving $\mathcal{X}_s$ once in $S$ is at most $p = \frac{M + L_V \cdot \Delta}{M + L_V \cdot \Delta + \delta} < 1$. Furthermore, once the stabilizing set $\mathcal{X}_s$ is left, we know that the value of $V$ is at most $\sup_{\mathbf{x} \in \mathcal{X}_s} V(\mathbf{x}) + L_V \cdot \Delta = \Gamma + L_V \cdot \Delta$ due to $L_V$ being the Lipschitz constant of $V$ and $\Delta$ being the maximum step size of the system. Thus, we have

$$\begin{aligned} \sup_{\mathbf{x} \in \mathcal{X}} \mathbb{E}_{\mathbf{x}}[\mathsf{OutAfter}_{\mathcal{X}_s}] &\leq p \cdot \Big( \sup_{\mathbf{x} \in \mathcal{X} \text{ s.t. } V(\mathbf{x}) \leq \Gamma + L_V \cdot \Delta} \mathbb{E}_{\mathbf{x}}[T_S] + \sup_{\mathbf{x} \in \mathcal{X}} \mathbb{E}_{\mathbf{x}}[\mathsf{OutAfter}_{\mathcal{X}_s}] \Big) \\ &\leq p \cdot \Big( \frac{\Gamma + L_V \cdot \Delta}{\epsilon} + \sup_{\mathbf{x} \in \mathcal{X}} \mathbb{E}_{\mathbf{x}}[\mathsf{OutAfter}_{\mathcal{X}_s}] \Big), \end{aligned}$$

where in the second inequality we again use the second item in Proposition 1 but now applied to the $\epsilon$-RSM $(X_i)_{i=0}^{\infty}$ with respect to the stopping time $T_S$ defined in the probability space of all system trajectories that start in the initial state $\mathbf{x}$. Hence, by deducting $p \cdot \sup_{\mathbf{x} \in \mathcal{X}} \mathbb{E}_{\mathbf{x}}[\mathsf{OutAfter}_{\mathcal{X}_s}]$ from both sides of the inequality and then dividing both sides of the resulting inequality by $1 - p > 0$, we conclude that

$$\sup_{\mathbf{x} \in \mathcal{X}} \mathbb{E}_{\mathbf{x}}[\mathsf{OutAfter}_{\mathcal{X}_s}] \leq \frac{p \cdot (\Gamma + L_V \cdot \Delta)}{(1 - p) \cdot \epsilon}.$$

Therefore, since $p = \frac{M + L_V \cdot \Delta}{M + L_V \cdot \Delta + \delta}$, we deduce that

$$\sup_{\mathbf{x} \in \mathcal{X}} \mathbb{E}_{\mathbf{x}}[\mathsf{OutAfter}_{\mathcal{X}_s}] \leq \frac{(M + L_V \cdot \Delta) \cdot (\Gamma + L_V \cdot \Delta)}{\delta \cdot \epsilon}. \tag{6}$$

By comgining eq. equation 4, equation 5 and equation 6, we deduce the first item in Theorem 2.

The second item in Theorem 2 follows immediately from the first item in Theorem 2 and an application of Markov's inequality which implies that $\mathbb{P}_{\mathbf{x}_0}[\mathsf{Out}_{\mathcal{X}_s} \geq t] \leq \frac{\mathbb{E}_{\mathbf{x}_0}[\mathsf{Out}_{\mathcal{X}_s}]}{t}$ for any $t > 0$. $\square$

## C  REGULARIZATION TERMS

Here, we provide details on the two regularization objectives that we add to the training loss.

**Global minimum regularization** We add the term $\mathcal{L}_{<M}(\theta, \nu)$ to the loss function, which is an auxiliary loss guiding the learner towards learning an sRSM candidate $V_\nu$ that attains the global minimum in the set $\{\mathbf{x} \in \mathcal{X} \mid V(\mathbf{x}) < M\}$. In particular, we impose a set $T \subseteq \mathcal{X}_s$ to have value $< M$ and the global minimum of the sRSM being in $T$. While this loss term does not enforce any

of the conditions in Definition 3 directly, we observe that it helps our learning process. It is defined via

$$\mathcal{L}_{<M}(\theta, \nu) = \max\{\max_{x_1,\dots x_{N_3} \in \mathcal{D}_{<M}} V_\nu(x) - M, 0\} + \max\{\min_{x_1,\dots x_{N_4} \in \mathcal{X}} V_\nu(x) - \min_{x_1,\dots x_{N_3} \in \mathcal{D}_{<M}} V_\nu(x), 0\}.$$

where $\mathcal{D}_{<M}$ is a set of states at which the sRSM canidate learned in the previous learning iteration is $< M$ and $N_3$ and $N_4$ are algorithm parameters.

**Lipschitz regularization** We regularize Lipschitz bounds of $V_\nu$ and $\pi_\theta$ during trainin by adding the regularization term

$$\lambda(\mathcal{L}_{\text{Lipschitz}}(\theta) + \mathcal{L}_{\text{Lipschitz}}(\nu)) + \alpha \mathcal{L}'_{\text{Lipschitz}}(\nu), \tag{7}$$

to the training objective, with

$$\mathcal{L}_{\text{Lipschitz}}(\phi) = \max\left\{ \prod_{W,b \in \phi} \max_j \sum_i |W_{i,j}| - \rho, 0 \right\}$$

and

$$\mathcal{L}'_{\text{Lipschitz}}(\phi) = \min\left\{ \prod_{W,b \in \phi} \max_j \sum_i |W_{i,j}| - \rho', 0 \right\}.$$

## D  PROOF OF THEOREM 3

**Theorem.** *Suppose that the verifier shows that $V_\nu$ satisfies eq. (2) for each $\widetilde{\mathbf{x}} \in \widetilde{\mathcal{X}}_{\geq M}$ and eq. (3) for each cell $\in Cells_{\mathcal{X} \setminus \mathcal{X}_s}$. Then $V_\nu$ is an sRSM and $\mathcal{X}_s$ is a.s. asymptotically stable under $\pi_\theta$.*

*Proof.* To prove the theorem, we first need to show that $V_\nu$ satisfies the three conditions in Definition 3.

Condition 1 in Definition 3 is satisfied by default since $V_\nu$ applies the softplus activation function to its output which ensures nonnegativity.

To deduce condition 2 in Definition 3, we need to show that there exists $\epsilon > 0$ such that for each $\mathbf{x} \in \mathcal{X}$ with $V_\nu(\mathbf{x}) \geq M$ we have

$$\mathbb{E}_{\omega \sim d}\left[V_\nu\left(f(\mathbf{x}, \pi(\mathbf{x}), \omega)\right)\right] \leq V(\mathbf{x}) - \epsilon.$$

We show that

$$\epsilon = \min_{\widetilde{\mathbf{x}} \in \widetilde{\mathcal{X}}_{\geq M}} \left(V(\widetilde{\mathbf{x}}) - \tau \cdot K - \mathbb{E}_{\omega \sim d}\left[V\left(f(\widetilde{\mathbf{x}}, \pi(\widetilde{\mathbf{x}}), \omega)\right)\right]\right)$$

satisfies this requirement. Fix $\mathbf{x} \in \mathcal{X}$ with $V_\nu(\mathbf{x}) \geq M$ and let $\widetilde{\mathbf{x}} \in \widetilde{\mathcal{X}}$ be such that $||\mathbf{x} - \widetilde{\mathbf{x}}||_1 \leq \tau$. Such $\widetilde{\mathbf{x}}$ exists by definition of a discretization. Furthremore, since $V_\nu(\mathbf{x}) \geq M$, the center of the cell that contains $\mathbf{x}$ must be contained in $\widetilde{\mathcal{X}}_{\geq M}$ so therefore we may pick such $\widetilde{\mathbf{x}} \in \widetilde{\mathcal{X}}_{\geq M}$ (the correctness of the computation of $\widetilde{\mathcal{X}}_{\geq M}$ follows from the correctness of IA-AI (Cousot & Cousot, 1977; Gowal et al., 2018)). Then, by Lipschitz continuity of $f$, $\pi_\theta$ and $V_\nu$, we have that

$$\begin{aligned}
&\mathbb{E}_{\omega \sim d}\left[V_\nu\left(f(\mathbf{x}, \pi_\theta(\mathbf{x}), \omega)\right)\right] \\
&\leq \mathbb{E}_{\omega \sim d}\left[V_\nu\left(f(\widetilde{\mathbf{x}}, \pi_\theta(\widetilde{\mathbf{x}}), \omega)\right)\right] + ||f(\widetilde{\mathbf{x}}, \pi_\theta(\widetilde{\mathbf{x}}), \omega) - f(\mathbf{x}, \pi(\mathbf{x}), \omega)||_1 \cdot L_V \\
&\leq \mathbb{E}_{\omega \sim d}\left[V_\nu\left(f(\widetilde{\mathbf{x}}, \pi_\theta(\widetilde{\mathbf{x}}), \omega)\right)\right] + ||(\widetilde{\mathbf{x}}, \pi_\theta(\widetilde{\mathbf{x}}), \omega) - (\mathbf{x}, \pi(\mathbf{x}), \omega)||_1 \cdot L_V \cdot L_f \\
&\leq \mathbb{E}_{\omega \sim d}\left[V_\nu\left(f(\widetilde{\mathbf{x}}, \pi_\theta(\widetilde{\mathbf{x}}), \omega)\right)\right] + ||\widetilde{\mathbf{x}} - \mathbf{x}||_1 \cdot L_V \cdot L_f \cdot (1 + L_\pi) \\
&\leq \mathbb{E}_{\omega \sim d}\left[V_\nu\left(f(\widetilde{\mathbf{x}}, \pi_\theta(\widetilde{\mathbf{x}}), \omega)\right)\right] + \tau \cdot L_V \cdot L_f \cdot (1 + L_\pi),
\end{aligned} \tag{8}$$

On the other hand, by Lipschitz continuity of $V_\nu$ we have

$$V_\nu(\mathbf{x}) \geq V_\nu(\widetilde{\mathbf{x}}) - ||\widetilde{\mathbf{x}} - \mathbf{x}||_1 \cdot L_V \geq V_\nu(\widetilde{\mathbf{x}}) - \tau \cdot L_V. \tag{9}$$

Thus combining eq.(8) and (9) we get that

$$
\begin{aligned}
V_\nu(\mathbf{x}) &- \mathbb{E}_{\omega \sim d}\Big[V_\nu\Big(f(\mathbf{x}, \pi_\theta(\mathbf{x}), \omega)\Big)\Big] \\
&\geq V_\nu(\widetilde{\mathbf{x}}) - \tau \cdot L_V - \mathbb{E}_{\omega \sim d}\Big[V_\nu\Big(f(\widetilde{\mathbf{x}}, \pi_\theta(\widetilde{\mathbf{x}}), \omega)\Big)\Big] - \tau \cdot L_V \cdot L_f \cdot (1 + L_\pi) \qquad (10) \\
&= V_\nu(\widetilde{\mathbf{x}}) - \tau \cdot K - \mathbb{E}_{\omega \sim d}\Big[V_\nu\Big(f(\widetilde{\mathbf{x}}, \pi_\theta(\widetilde{\mathbf{x}}), \omega)\Big)\Big] \geq \epsilon,
\end{aligned}
$$

The last inequality holds by our definition of $\epsilon$, therefore we conclude that $V_\nu$ satisfies condition 2 in Definition 3.

Finally, to deduce condition 3 in Definition 3, we need to show that there exists $\delta > 0$ such that $V_\nu(\mathbf{x}) \geq M + L_V \cdot \Delta + \delta$ holds for each $\mathbf{x} \in \mathcal{X} \backslash \mathcal{X}_s$. But the fact that

$$
\delta = \min_{\text{cell} \in \text{Cells}_{\mathcal{X} \backslash \mathcal{X}_s}} \{\underline{V}_\nu(\text{cell}) - M - L_V \cdot \Delta_\theta\}
$$

satisfies the claim follows immediately from correctness of IA-AI and the fact that eq. (3) holds for each cell $\in \text{Cells}_{\mathcal{X} \backslash \mathcal{X}_s}$.

Thus, this concludes the proof that $V_\nu$ satisfies the three conditions in Definition 3. Then, by Theorem 1 on sRSMs, we know that $\mathcal{X}_s$ is a.s. asymptotically stable under $\pi_\theta$. $\qquad \square$

# E  EXPERIMENTAL EVALUATION DETAILS

We implemented our algorithm in JAX. All experiments were run on a 4 CPU-core machine with 64GB of memory and an NVIDIA A10 with 24GB of memory.

**Benchmark environments** The dynamics of the two-dimensional dynamical system (2D system) are defined as

$$
\mathbf{x}_{t+1} = \begin{pmatrix} 1 & 0.0196 \\ 0 & 0.98 \end{pmatrix} \mathbf{x}_t + \begin{pmatrix} 0.002 \\ 0.1 \end{pmatrix} g(\mathbf{u}_t) + \begin{pmatrix} 0.002 & 0 \\ 0 & 0.001 \end{pmatrix} \omega, \qquad (11)
$$

where $\omega$ is a disturbance vector and $\omega[1], \omega[2] \sim$ Triangular. The function $g$ bounds the range of admissible actions by $g(u) = \max(\min(u, 1), -1)$.

The probability density function of Triangular is defined by

$$
\text{Triangular}(x) := \begin{cases} 0 & \text{if } x < -1 \\ 1 - |x| & \text{if } -1 \leq x \leq 1 \\ 0 & \text{otherwise} \end{cases} . \qquad (12)
$$

The dynamics function of the inverted pendulum task is defined as

$$
\begin{aligned}
\mathbf{x}_{t+1}[2] &:= (1 - b)\mathbf{x}_t[2] \\
&+ d \cdot \Big(\frac{-1.5 \cdot G \cdot \sin(\mathbf{x}_t[1] + \pi)}{2l} + \frac{3}{ml^2} 2g(\mathbf{u}_t)\Big) \\
&+ 0.002\omega[1] \\
\mathbf{x}_{t+1}[1] &:= \mathbf{x}_t[1] + d \cdot \mathbf{x}_{t+1}[2] + 0.005\omega[2],
\end{aligned}
$$

where the parameters $d, G, m, l, b$ are defined in Table 2. For training a policy on the inverted pendulum task, we used a reward $r_t$ at time $t$ defined by $r_t := 1 - \mathbf{x}_t[1]^2 - 0.1\mathbf{x}_t[2]^2$.

The hyperparameters we used in the experiments for learning the policy and the sRSM are listed in Table 3. For each of the tasks, we consider $T = \{x \mid |x_1| \leq 0.2, |x_2| \leq 0.2\}$.

We observed a better convergence and more stable training when training only the sRSM candidate and keep the weights of the policy frozen for the first three iterations of our algorithm. For the second task we replaced $\epsilon_{\text{train}}$ with $K_{\theta,\nu} \cdot \tau$ during the training. Specifically, instead of using $L_{\text{cond 2}}(\theta, \nu)$, we set

| Parameter | Value |
|:---:|:---:|
| $d$ | 0.05 |
| $G$ | 10 |
| $m$ | 0.15 |
| $l$ | 0.5 |
| $b$ | 0.1 |

Table 2: Parameters of the inverted pendulum task.

| Parameter | Value |
|:---:|:---:|
| Learning rate | 0.0005 |
| $\lambda$ | 0.001 |
| $\alpha$ | 10 |
| $\rho_\theta$ | 4 |
| $\rho_\nu$ | 8 |
| $\rho'$ | 0.01 |
| $\delta_{\text{train}}$ | 0.1 |
| $N_{\text{cond 2}}$ | 16 |
| $N_{\text{cond 3}}$ | 256 |
| $N_3$ | 256 |
| $N_4$ | 512 |
| $\epsilon_{\text{train}}$ | 0.1 |

Table 3: Hyperparameters used in our experiments.

$$\mathcal{L}'_{\text{cond 2}}(\theta, \nu) = \frac{1}{|B|} \sum_{\mathbf{x} \in B} \left( \max \left\{ \sum_{\omega_1, \ldots, \omega_{N_{\text{cond 2}}} \sim d} \frac{V_\nu\big(f(\mathbf{x}, \pi_\theta(\mathbf{x}), \omega_i)\big)}{N_{\text{cond 2}}} - V_\nu(\mathbf{x}) + K_{\theta, \nu} \cdot \tau, 0 \right\} \right).$$

For the inverted pendulum task, the plots and the results in Table 1 in the main paper are obtained by training with $\mathcal{L}'_{\text{cond 2}}(\theta, \nu)$ as the loss function. Here, we performed an ablation study to test whether using $\mathcal{L}'_{\text{cond 2}}(\theta, \nu)$ can improve the results, i.e., whether the number of iterations is decreased. The results in Table 3 show that the effectiveness of using $\mathcal{L}'_{\text{cond 2}}(\theta, \nu)$ on the particular system.

**Grid refinement**

We implemented two types of grid refinement procedures to refine the mesh of the discretization used by the verifier. The first refinement is scheduled to multiply $\tau$ by 0.5 every second iteration starting at iteration 5 if no *hard violation* is encountered by the verifier module. A violation is a counterexample to condition 2 in Definition 3 in the main paper. Hard violations are violations that also violate the condition

$$\mathbb{E}_{\omega \sim d}\Big[V\Big(f(\mathbf{x}, \pi(\mathbf{x}), \omega)\Big)\Big] < V(\mathbf{x}).$$

| Environment | Use $\mathcal{L}'_{\text{cond 2}}(\theta, \nu)$ | Iterations | Mesh ($\tau$) | $p$ | Runtime |
|:---:|:---:|:---:|:---:|:---:|:---:|
| 2D system | No | 5 | 0.0007 | 0.80 | 3660 s |
| | Yes | 7 | 0.0007 | 0.78 | 4405 s |
| Inverted pendulum | No | 8 | 0.003 | 0.97 | 7004 s |
| | Yes | 4 | 0.003 | 0.97 | 2619 s |

Table 4: Ablation analysis of the impact of the loss term $\mathcal{L}'_{\text{cond 2}}(\theta, \nu)$. Number of learner-verifier loop iterations, mesh of the discretization used by the verifier, $p$, and total algorithm runtime (in seconds).

Our second refinement procedure is invoked when there are violations but no hard violations. In this case, our procedure tries to verify grid cells where violations were observed using a mesh of $0.5\tau$.

### E.1 PPO DETAILS

The settings used for the PPO Schulman et al. (2017) pre-training process are as follows. In each PPO iteration, 30 episodes of the environment are collected in a training buffer. Stochastic is introduced to the sampling of the policy network $\pi_\mu$ using a Gaussian distributed random variable added to the policy's output, i.e., the policy predicts a Gaussian's mean. The standard deviation of the Gaussian is dynamic during the policy training process according to a linear decay starting from 0.5 at first PPO iteration to 0.05 at PPO iteration 50. The advantage values are normalized by subtracting the mean and scaling by the inverse of the standard deviation of the advantage values of the training buffer. The PPO clipping value $\varepsilon$ is 0.2 and $\gamma$ is set to 0.99. In each PPO iteration, we train the policy for 10 epochs, except for the first iteration where we train the policy for 30 epochs. An epoch accounts to a pass over the entire data in the training buffer, i.e., the data from the the rollout episodes. We train the value network 5 epochs, expect in the first PPO iteration, where we train the value network for 10 epochs. The Lipschitz regularization is applied to the learning of the policy parameters during the PPO pre-training.

